# Transitional Channel Flow: A Minimal Stochastic Model

**DOI:** 10.3390/e22121348

**Published:** 2020-11-29

**Authors:** Paul Manneville, Masaki Shimizu

**Affiliations:** 1LadHyX, École Polytechnique, CNRS, Institut Polytechnique de Paris, 91128 Palaiseau, France; 2Graduate School of Engineering Science, Osaka University, Toyonaka 560-0043, Japan; shimizu@me.es.osaka-u.ac.jp

**Keywords:** transition to/from turbulence, wall-bounded shear flow, plane Poiseuille flow, spatiotemporal intermittency, directed percolation, critical phenomena

## Abstract

In line with Pomeau’s conjecture about the relevance of directed percolation (DP) to turbulence onset/decay in wall-bounded flows, we propose a minimal stochastic model dedicated to the interpretation of the spatially intermittent regimes observed in channel flow before its return to laminar flow. Numerical simulations show that a regime with bands obliquely drifting in two stream-wise symmetrical directions bifurcates into an asymmetrical regime, before ultimately decaying to laminar flow. The model is expressed in terms of a probabilistic cellular automaton of evolving von Neumann neighborhoods with probabilities educed from a close examination of simulation results. It implements band propagation and the two main local processes: longitudinal splitting involving bands with the same orientation, and transversal splitting giving birth to a daughter band with an orientation opposite to that of its mother. The ultimate decay stage observed to display one-dimensional DP properties in a two-dimensional geometry is interpreted as resulting from the irrelevance of lateral spreading in the single-orientation regime. The model also reproduces the bifurcation restoring the symmetry upon variation of the probability attached to transversal splitting, which opens the way to a study of the critical properties of that bifurcation, in analogy with thermodynamic phase transitions.

## 1. Context

How laminar flow becomes turbulent, or the reverse, when the shearing rate changes, is a problem of great conceptual interest and practical importance. This special issue is focused on the case when the transition is characterized by the fluctuating coexistence of domains either laminar or turbulent in physical space at a given Reynolds number Re (control parameter), a regime called spatiotemporal intermittency, relevant to wall-bounded flows in particular. Several years ago, Y. Pomeau [1] placed that problem in the realm of statistical physics by proposing its approach in terms of a non-equilibrium phase transition called directed percolation (DP). This process displays specific statistical properties defining a universality class liable to characterize systems with two competing local states, one active, the other absorbing, with remarkably simple dynamical rules: any active site may contaminate a neighbor and/or decay into the absorbing state, and an absorbing state cannot give rise to any activity [2]. The coexistence is regulated by the contamination probability, and a critical point can be defined above which the mixture of active and absorbing states is sustained and below which the active state recedes, leaving room for a globally absorbing state. The fraction of active sites is a measure of the global status of the system. The subcritical context typical of wall-bounded flows, initially pointed out by Pomeau, seems an interesting testbed for universality [3,4]. Here, turbulence plays the role of the active state and laminar flow, being linearly stable, represents the absorbing state. DP has indeed been shown relevant to simple shear between parallel plates (Couette flow) [5] and its stress-free version (Waleffe flow) [6]. The most recent contributions to the field can be found in [7]. In this paper we will be interested in plane channel flow (also called plane Poiseuille flow), the flow driven by a pressure gradient between two parallel plane plates, which is not fully understood despite recent advances.

In this context, universal properties are notably difficult to extract from experiments, since they relate to the thermodynamic limits of asymptotically large systems in the long time limit, whereas what plays the role of microscopic scales involves already macroscopic agents, e.g., roll structures in convection or turbulent streaks in open flows, and the turnover time associated with such structures. However, universality focuses on quantitative aspects of systems sharing the same qualitative characteristics, in particular symmetries and the effective space dimension *D* in which these systems evolve. Delicate questions can thus be attacked by modeling attempts that implement these traits appropriately. This approach involves simplifications from the primitive equations governing the problem, here the Navier–Stokes equations, to low-order differential models implementing the building blocks of the dynamics [8], to coupled map lattices (CML) in which the evolution is rendered by maps and space is discretized [9,10], to cellular automata for which local state variables are also discretized, and ultimately to probabilistic cellular automata (PCA), where the evolution rule itself becomes stochastic [11]. The absence of a rigorous theoretical method supporting the passage from one modeling level to the next, such as multi-scale expansions or Galerkin approximations, makes the simplification rely on careful empirical observations of the case under study, which somehow comes and limits the breadth of the conclusions drawn.

### 1.1. Physical Context: Plane Channel Flow

Of interest here, the transitional range of plane channel flow displays a remarkable series of steps at decreasing Re from large values where a regime of featureless turbulence prevails. It has been the subject of numerous studies and references to them can be found in the article by Kashyap, Duguet, and Dauchot in this special issue [12]; see also [13]. Our own observations based on numerical simulations are described in [14,15] and summarized in Figure 1.

The Reynolds number used to characterize the flow regime is defined as Re=Uch/ν, where 2h is the gap between the plates, $U_{c}$ is the mid-gap stream-wise speed of a supposedly laminar flow under the considered pressure gradient, and ν the kinematic viscosity. This definition using $U_{c}$ is appropriate for our numerical simulations under constant pressure-gradient driving. Other definitions involve the friction velocity $U_{\tau }$, or the stream-wise speed averaged over the gap $U_{b}$. They are related either empirically, vis., $U_{b}$ vs. $U_{c}$, or theoretically, vis., Reτ=2Re to be used in particular for connecting to the work presented in [12], and some other articles. See [14] for details. Below a first threshold Ret, featureless turbulence leaves room for a laminar–turbulent, oblique, patterned regime (upper transitional range) that next turns into a sparse arrangement of localized turbulent bands (LTBs) propagating obliquely along two directions symmetrical with respect to the general stream-wise flow direction, experiencing collisions and splittings (“two-sided” lower transitional regime). Event B in Figure 1 corresponds to the opening of laminar gaps along the intertwined band arrangement observed in the tight laminar–turbulent network regime, and the simultaneous prevalence of downstream active heads (DAHs) driving the LTBs. Upon decreasing Re further, a symmetry-breaking bifurcation takes place at a second threshold Re2, below which a single LTB orientation prevails. Figure 2 displays snapshots of the flow illustrating these last two stages.

A significant result in [14] was that the decrease of turbulence intensity with Re below event *B* followed expectations for directed percolation in two dimensions but that, controlled by the decreasing probability of transversal splitting, the bifurcation at Re2 prevented the flow to reach the corresponding threshold. The latter could nevertheless be extrapolated to a value ReDP<Re2. The ultimate decay stage takes place at Reynolds numbers below the point whereat transversal splitting ceases to operate. Figure 3 illustrates an extremely rare occurrence of transversal splitting at a Reynolds number roughly corresponding to event A in Figure 1.

At lower Re, deprived of the possibility to nucleate daughters’ LTBs of opposite propagation orientation, LTBs are forcibly maintained in the “one-sided” regime that eventually decays below a third threshold Reg, marking the global stability of the laminar flow. Corresponding flow patterns are illustrated in Figure 4, the right panel of which displays the surprising result that the turbulent fraction decreases as a power law with an exponent β of the order of that for directed percolation in one dimension, despite the fact that the flow develops in two dimensions [16].

The objective of the present work is the design of a minimal PCA model for these two last stages that is applicable to flow states for Re below event B, incorporates the anisotropy features visible in Figure 2, Figure 3 and Figure 4, and accounts for the specific role transversal splitting above event A, in view of providing clues to their statistical properties in relation to dimensionality and universality issues.

### 1.2. Modeling Context: Directed Percolation, Probabilistic Cellular Automata, and Criticality Issues

Various modeling approaches to transitional wall-bounded flows have received considerable attention recently, from low-order Galerkin expansions of the primitive equations [17,18], to phenomenological theories based on a deep physical analysis of the processes involved in a reaction-diffusion context [19], to analogical systems expressed in terms of deterministic coupled map lattices [6,10], and to more conceptual models implementing the dynamics of cellular automata with probabilistic evolution rules (PCA) [20,21,22]. The model developed below belongs to this last category, implementing rules that focus on the main qualitative features seen in experiments. Such models are based on the conventional modeling of DP [2] which is most appropriate to account for the absorbing versus active character of local states.

Let us briefly recall the PCA/DP framework. In the most general case, the activity at site *j* at time t+1, call it $S_{j}\in \left\{0,1\right\}$, depends on the activity at sites in a full *D*-dimensional neighbor $V_{j}$ of that site at time *t* and the status of the links, permitting or not the transfer of activity within the neighborhood. For convenience a (D+1)-dimensional lattice is defined with one-way (directed) bonds in the direction corresponding to time so that *D*-dimensional directed percolation is often presented as a special (D+1)-dimensional percolation problem. In the simplest case of one space dimension (D=1), the neighborhood of a lattice site at *j* is the set of sites with $j^{\prime }\in \left[j-r_{1},j+r_{2}\right]$, comprising $r_{2}+r_{1}+1$ sites, and it is supposed that contamination of the state at *j* at time t+1 depends on the status of full configuration, the sites’ activity, and the bonds’ transfer properties (“bond–site” percolation [23]). In some systems, the propagation rule is totalistic in the sense that the output only depends on the number of active sites in the neighborhood and not on their positions, i.e., $\varsigma_{j}=\sum_{j^{\prime }\in V_{j}}S_{j^{\prime }}$; an interesting example is given in [24].

In view of future developments, let us discuss bond directed percolation in one dimension (D=1) with two neighbors ($r_{1}=0$ or $r_{2}=0$), only depending on the probability *p* that bonds transfer activity. The evolution rule $S_{j}^{\prime }=R\left(S_{j},S_{j+1}\right)$, where $S_{j}^{\prime }$ denotes the state at node *j* and time t+1, is totalistic. With $\varsigma_{j}=S_{j}+S_{j+1}$, we have (a) R(ς=0)=0 with probability 1 (a site connected to two absorbing parents never gets active whatever the links) and (b) R(ς=1)=1 with probability *p* (closed link transmitting activity), so that (a’) R(ς=1)=0 with probability 1−p (open link preventing transmission), (c) R(ς=2)=0 with probability ${\left(1-p\right)}^{2}$ (absorbing since the two links are open), and (d) R(ς=2)=1 with probability $1-{\left(1-p\right)}^{2}=p\left(2-p\right)$, the complementary case.

The question is whether, depending on the value of *p*, once initiated, activity keeps continuing in the thermodynamic limit of infinite times in an infinitely wide system. An answer is readily obtained in the mean-field approximation where actual local states are replaced by their mean value, neglecting the effect of spatial correlations and stochastic fluctuations (we follow the presentation of [20]). The spatially-discrete Boolean variables $S_{j}$ are, therefore, replaced by their spatial averages $S=\langle S_{j}\left(t\right)\rangle$ and this mean value is just the probability that any given site is active. It is then argued that the probability to get a future absorbing state, $1-S^{\prime }$, is given by activity not being transmitted ${\left(1-pS\right)}^{2}$, which yields the mean-field equation:(1)$$1-S^{\prime }={\left(1-pS\right)}^{2}=1-2pS+p^{2}S^{2},\;i.e.,\;S^{\prime }=2pS-p^{2}S^{2}.$$
Equilibrium states correspond to the fixed points of (Equation 1): $S^{\prime }=S=S*$, which gives a nontrivial activity level $S_{*}=\left(2p-1\right)/p^{2}$ when $p\geq p_{c}=1/2$. Close to threshold, defining $\varepsilon =\left(p-p_{c}\right)/p_{c}=2p-1$ one gets $S_{*}\approx 4\varepsilon$. In the mean-field (MF) approximation $S_{*}$ is the order parameter of the transition supposed to vary as $\varepsilon^{\beta }$, which defines the critical exponent β, here $\beta_{\mathrm{MF}}=1$. Directed percolation is the prototype of non-equilibrium phase transitions and, as such, is associated with a set of critical exponents (see [2]). Both the critical probability $p_{c}$ and the mean activity $S_{*}$ are affected by the effects of fluctuations, with $p_{c}\approx 0.6445>1/2$ expressing that a probability larger than the mean-field estimate is necessary to preserve activity, and $\beta_{\mathrm{DP}}\approx 0.276$ when D=1. The simple mean-field argument is not sensitive to the value of *D* in contrast with reality: $\beta_{\mathrm{DP}}\approx 0.584$ when D=2, ≈0.81 when D=3, and trends upwards to 1 reached at $D=4=D_{c}=4$, called the upper critical dimension (see [2] for a review). Quite generally, mean-field arguments are valid for $D>D_{c}$. We are interested in another critical exponent, α. When starting from a fully active system exactly poised at $p_{c}$, the turbulent fraction is observed to decrease with time (the number of iteration steps) as $\langle S\rangle \propto t^{-\alpha }$ with α≈0.159 when D=1 and 0.451 when D=2, whereas the mean-field prediction, easily derived from (Equation 1), is $\alpha_{\mathrm{MF}}=1$. Scaling theory shows that $\alpha =\beta /\nu_{\|}$, where $\nu_{\|}$ is the exponent accounting for the decay of time correlations while $\nu_{⊥}$ describes the decay of space correlations [2].

Universality is a key concept in the field of critical phenomena characterizing continuous phase transitions. It leads to the definition of universality classes expressing the insensitivity of critical properties to specific characteristics of the systems and retaining only properties linked to the symmetries of the order parameter and the dimension of space. For directed percolation, universality is conjectured to be ruled by a few conditions put forward by Grassberger and Janssen: that the transition is continuous into a unique absorbing state and characterized by a positive one-component order parameter, and that the processes involved are short-range and without weird properties such as quenched randomness; see [2]. Universality issues are discussed at length elsewhere in this special issue, in particular by Takeda et al. [25].

In this first approach, we shall examine how universality expectations hold for the ultimate decay stage of transitional channel flow at Reg, as described in Section 1.1, and limit the discussion to the consideration of exponents β and α. This will be done in Section 3, the next section being devoted to the derivation of the model and its mean-field study. Section 4 focuses on its ability to account for the symmetry-breaking bifurcation at Re2, and our conclusions are presented in Section 5.

## 2. Description of the Model

### 2.1. Context

The approach to be developed is not new in the field of transitional flows. For example, studying plane channel flow, Sano and Tamai [21] introduced a plain 2D-DP model dedicated to support their experimental results, with a simple spatial shift implementing advection and a uniformly turbulent state upstream corresponding to their setup. Earlier, a similarly conceptual model was examined by Allhoff and Eckhardt [20], who introduced a PCA with two parameters accounting for persistence and lateral spreading appropriate for the symmetries of plane Couette flow, developed its mean-field treatment, and performed simulations to illustrate the spreading of spots and decay of turbulence in agreement with expectations. In a similar spirit but introducing more physical input, Kreilos et al. [22] analyzed the development of turbulent spots in boundary layers as a function of the residual turbulence level upstreams, separating a deterministic transport step from a stochastic growth/decay step with probabilities extracted from a numerical experiment, gaining insight into the statistics of boundary layer receptivity.

Following the lines of research suggested by those works, we developed a 2D model designed to interpret the decay of channel in the LTB regimes from two-sided to one-sided at decreasing Re, just qualitatively proposing a plausible variation of probabilities introduced as functions of Re. In our approach, the elementary agents are the LTBs themselves either propagating to the left or to the right of the stream-wise direction. To them we attach variables analogous to spins in magnetic phase transitions problems. Even if in computations, numerical values S=±1 will be used, for descriptive and graphical convenience we shall associate them with colors—specifically: blue (*B*) and red (*R*) for right- and left-propagating LTBs, respectively. Laminar sites will be denoted using the empty-set symbol ∅, will have value 0, and will be graphically left blank. These agents will be seated at the nodes of a square lattice with coordinates (i,j), i.e., $S_{\left(i,j\right)}$ with S↦{R,B,∅} at the given site. As seen in Figure 5a, we place the stream-wise direction along the first diagonal of the lattice so that the LTBs will move along the horizontal and vertical axes; see Figure 5b.

A strong assumption is that an LTB as a whole corresponds to a single active state, while the discretization of space coordinates $\left(i,j\right)\in Z^{2}$, and time t∈N tacitly refers to an appropriate rescaling of time and space. Furthermore, interactions are taken as local, with configurations limited to nearest neighbors in each space direction. Accordingly, the dynamics at a site (i,j) only depend on the configuration of its von Neumann neighborhood $V_{\left(i,j\right)}:=\left\{\left(i,j\right),\left(i\pm 1,j\right),\left(i,j\pm 1\right)\right\}$, Figure 5c, while evolution is driven by a random process. We now turn to the definition of rules that mimic the actual continuous space-time, subcritical and chaotic, Navier–Stokes dynamics governing the LTBs’ propagation, decay, splitting, and collisions, via educated guesses from the scrutiny of simulation results, in particular those in the supplementary material attached to [14].

### 2.2. Design of the Model

Let us first give a brief description of the processes to be accounted for. Below Re≈800 (event A) only decay and longitudinal splittings are possible. Not visible in the snapshots of Figure 4 (left) but observable in the movies is the fact that a daughter LTB resulting from longitudinal splitting runs behind its mother along a track that may be slightly shifted upstream. This shift is negligible when Re is small (in-line longitudinal splitting) but as Re increases it becomes more and more visible while the general propagation direction is unchanged (off-aligned longitudinal splitting). On the other hand, Figure 4 clearly illustrates the fact that, upon transversal splitting, the new-born LTB systematically develops on the downstream side of its parent. Importantly, the propagation of LTBs is a dynamical feature different from advection treated as a deterministic step in [22]. Accordingly, it will be understood as a statistical propensity to move in a given direction resulting from an imbalance of stochastic “forward” and “backward” processes along their directions of motion. Other complex processes also seen in the simulations, such as fluctuating propagation with acceleration, slowing down, or lateral wandering, will be included only in so far as they can be decomposed into such more elementary events. All the events to be included in the model can be translated into the language of reaction–diffusion processes, persistence or death, offspring production, and coalescence, common in the field of DP theory [2].

On general grounds the governing equation reads:(2)$$S^{\prime }\left(i,j\right)=\sum_{C^{\prime }}R_{C^{\prime }}\delta_{C^{\prime }C\left(i,j\right)}\;,$$
where C(i,j) is the neighborhood configuration of site (i,j) at time *t*, $C^{\prime }$ one of the possible configurations, and $R_{C^{\prime }}$ a stochastic variable taking value 1 with probability $p_{C^{\prime }}$ corresponding to configuration $C^{\prime }$ and value 0 with probability $1-p_{C^{\prime }}$. The Kronecker symbol $\delta_{C^{\prime }C}$ is here to select the configuration $C^{\prime }$ that matches C. Depending on C and $C^{\prime }$, the output $S^{\prime }\left(i,j\right)$ can be *B* or *R*.

Figure 6 illustrates the set of possible single-colored neighborhoods, either *B* (upper line) or *R* (lower line). Following the indexation in Figure 5c, the order of the columns is based on the physical condition and respects the upstream/downstream distinction illustrated in Figure 5a, making configurations with the same index physically equivalent.

These single-color elementary configurations will be denoted as $C_{i}$ with i∈ [1:5]. They will be described as [SSSSS] with S=B, *R*, or ∅. Hence $C_{3}\equiv \left[\emptyset \;\emptyset \;B\;\emptyset \;\emptyset \right]$ or [∅R∅∅∅]. Later, more complicated configurations will not be given a name but just a description following the same rule, e.g., [∅BBR∅].

Importantly, we make the assumption that the future state at a given node, the question marks in Figure 6, is the result of the probabilistic combination of the independent contributions of elementary configurations involving a single active state in its neighborhood.

First of all, the void configuration $C_{0}\equiv \left[\emptyset \;\emptyset \;\emptyset \;\emptyset \;\emptyset \right]$ obviously generates an empty site with probability 1, hence an occupied site with probability $p_{C_{0}}=0$, in order to preserve the absorbing character of the dynamics. All the other configurations evolves according to probabilities that are free parameters just constrained by empirical observations. Let us now interpret probabilities associated with the five situations depicted in Figure 6, focusing on the case of *B* states:1.$C_{5}\equiv \left[\emptyset \;\emptyset \;\emptyset \;\emptyset \;B\right]$ corresponds to the natural propagation of the active state along its own motion direction. Accordingly, the active site *B* at (i−1,j) is expected to be found at (i,j) and time t+1 with a high probability, $p_{C_{5}}=p_{5}≲1$, which corresponds to the near-deterministic propagation of an active state as observed for Re≥Reg. With probability $1-p_{5}\ll 1$, site (i,j) will not turn active, which means that the LTB has decayed or experienced a speed fluctuation that delayed its propagation. The corresponding *R* configuration is $C_{5}\equiv \left[\emptyset \;\emptyset \;\emptyset \;R\;\emptyset \right]$.2.Configuration $C_{1}\equiv \left[B\;\emptyset \;\emptyset \;\emptyset \;\emptyset \right]$ corresponds to an active site *B* at (i,j) that is not supposed to stay in place but move to (i+1,j) with probability $p_{5}$ and leave site (i,j) empty at time t+1. The probability $p_{1}$ that (i,j) is still active at time t+1, therefore, generally corresponds to the creation of a novel active state by in-line longitudinal splitting at the rear of the active state that has effectively moved. Persisting activity at (i,j) and time t+1 can also be the result of state at (i,j) and time *t* experiencing a speed fluctuation leaving it stuck at the same place with probability $1-p_{5}$ as argued above for configuration $C_{5}$. The presence of parameter $p_{1}$ undoubtedly makes the dynamics richer. The corresponding *R*-configuration is $C_{1}\equiv \left[R\;\emptyset \;\emptyset \;\emptyset \;\emptyset \right]$.3.Configuration $C_{2}\equiv \left[\emptyset B\;\emptyset \;\emptyset \;\emptyset \right]$ corresponds to an active state *B* at site (i,j+1) that contaminates backwards and laterally upstream the site at (i,j) in addition to its likely propagation to (i+1,j+1) with probability $p_{5}$. This is precisely what is sometimes observed for longitudinal splitting, where the daughter follows a track parallel to that of the mother but slightly shifted upstream, i.e., off-aligned. Configurations $C_{1}$ and $C_{2}$ both account for longitudinal splitting but the latter hence introduces some lateral diffusion. Along this line of thought, numerical simulation results in [14], illustrated in Figure 4, suggest that probability $p_{2}$ is tiny close to Reg but increases with Re. The corresponding *R*-configuration is $C_{2}\equiv \left[\emptyset \;\emptyset R\;\emptyset \;\emptyset \right]$.4.In configuration $C_{3}\equiv \left[\emptyset \;\emptyset \;B\;\emptyset \;\emptyset \right]$, the active site *B* at (i+1,j) is supposed to advance further at (i+2,j) with probability $p_{5}$. Persisting activity at (i,j), therefore, means longitudinal splitting ahead but now with the opening of a wide laminar gap between the offspring left behind at (i,j) and the parent that has advanced, with probability $p_{5}$, at (i+2,j). Else, activity at (i,j) and t+1 could result from activity at (i+1,j) and time *t* propagating backwards to (i,j) at time t+1. These circumstances have not been observed and appears unlikely or impossible, which suggests to take $p_{3}=0$. The corresponding *R*-configuration is $C_{3}\equiv \left[\emptyset \;R\;\emptyset \;\emptyset \;\emptyset \right]$.5.In configuration $C_{4}\equiv \left[\emptyset \;\emptyset \;\emptyset \;B\;\emptyset \right]$, the state *B* at (i,j−1) and time *t* is expected to be at (i+1,j−1) at time t+1. State at (i,j) being active at t+1 means contamination backwards and laterally downstream, which is never observed in the simulations; hence, $p_{4}=0$. The corresponding *R*-configuration is $C_{4}\equiv \left[\emptyset \;\emptyset \;\emptyset \;\emptyset \;R\right]$.6.Still about configuration $C_{4}$, the situations described in the previous items all imply single-colored evolution, which is guaranteed below the onset of transversal splitting, i.e., R≲800. When Re≳800, as illustrated in Figure 3, this splitting produces an *R* offspring at (i,j) out of a *B* parent at (i,j−1) or *B* offspring from an *R* parent at (i−1,j), as sketched in Figure 7 (left). A probability $p_{4}^{\prime }\neq 0$ will be associated with it, where the prime is meant to recall that it involves states of different colors.

To summarize, as it stands the model involves four parameters: $p_{1}$ mainly governs longitudinal splitting and $p_{2}$ additional lateral diffusion, $p_{5}$ is for propagation, and $p_{4}^{\prime }$ for transversal splitting. The propagation of active states along their own direction involves probabilities associated with elementary configurations $C_{1}$ and $C_{5}$ while the overwhelming contribution of $p_{5}$ favors one direction. Configuration $C_{3}$ that could have contributed to the balance is empirically found negligible, saving one parameter as indicated above.

Neighborhoods with more than one active site are treated by assuming that the future state $S^{\prime }$ of the central node (i,j) is the combined output of its elementary ingredients, each contribution being considered as independent of the others, i.e., without memory of the anterior evolution, of which the considered configuration is the outcome. The computation of the probability attached to the output of a given single-colored neighborhood is then straightforward. The argument follows the lines given for directed percolation, bearing on the probability that the state at the node will be absorbing (empty) and leading to Equation (Equation 1) in the mean-field approximation [20,24]. Things are a little more complicated when the neighborhood is two-colored since in all mixed-colored cases some configurations correspond to collisions and others allow for the nucleation of a differently colored offspring when $p_{4}^{\prime }\neq 0$.

For an elementary configuration, non-contamination of site (i,j) from an active neighboring state in position k∈ [1:5] takes place with probability $\left(1-p_{k}\right)$ and of course with probability 1 if the corresponding site is empty. This gives the general formula $\left(1-p_{k}S_{k}\right)$, where $S_{k}=1$, when the site is active, either *B* or *R*, and $S_{k}=0$ when it is absorbing (∅). For a configuration $C_{x}=\left[S_{1},S_{2},S_{3},S_{4},S_{5}\right]$, where S=B, *R*, or ∅, the probability to get an absorbing state is $\left(1-p_{C_{x}}\right)=\prod_{k}\left(1-p_{k}S_{k}\right)$ hence for the node to be activated $p_{C_{x}}=1-\prod_{k}\left(1-p_{k}S_{k}\right)$. To deal with two-colored neighborhoods properly, we must be a little more specific and write the probability of the state $S^{\prime }$ of a given color *S* as
(3)$$p_{\left[S_{1},S_{2},{\bar{S}}_{4},S_{5}\right]}=1-\left(1-p_{1}S_{1}\right)\left(1-p_{2}S_{2}\right)\left(1-p_{4}^{\prime }{\bar{S}}_{4}\right)\left(1-p_{5}S_{5}\right)$$
where it is understood that if S=B, then $\bar{S}=R$ or the reverse, and $S_{j}=0$ for j=1,2,5, or ${\bar{S}}_{4}=0$ if the corresponding states are ∅. Figure 7 (right) illustrates the most interesting two-state configurations with different colors corresponding to collisions (C1) and offspring generation (C2). Such a situation is dealt with by adding a supplementary rule:

7.When the general expression (Equation 3) gives non-zero probabilities to $S^{\prime }$ and ${\bar{S}}^{\prime }$ the resulting superposition of states is not allowed and a choice has to be made. It might seem natural to keep the state with the maximum probability but, depending on circumstances hard to decipher, collisions sometimes appear to cause the decay of both protagonists or else reinforce the dominance of one color in a given region of space. A similar bias can affect transversal splitting. These peculiarities are not taken into account here: for simplicity, in all conflicting cases, we make the assumption that the result is non-empty and random with probability 1/2.

The model is now complete with parameters clearly related to empirical observations, plausible relative orders of magnitude and sense of variation: Probability $p_{5}$ is the main ingredient for the built-in propagation of the two families of LTBs (active states). In turn $p_{1}$ is obviously related to the behavior of the system close to decay at and slightly above Reg. The value given to probability $p_{2}$ will appear crucial to the 1D reduction of DP in a 2D medium as observed experimentally (Figure 4, right). Finally, we can anticipate that probability $p_{4}^{\prime }$ will control the one-sided/two-sided symmetry-restoring bifurcation, as it continuously grows from 0 beyond Event A at R≈800.

### 2.3. Mean-Field Approach

The explanatory potential of the model is first examined by means of a mean-field approximation which mainly relies on the replacement of fluctuating quantities by space-averaged values and the neglect of correlations. The observables involved in the mean-field expressions are the ensemble averages of the microscopic states 〈S(i,j)〉. Their values at t+1 are obtained by taking averages of the governing Equation (Equation 2) using the expression of the configurational probabilities given in (Equation 3). By assumption/definition $\langle S^{\prime }\rangle$ is the mean outcome of $p_{C_{x}}$ averaged over all the possible configurations, where space dependence (i,j) is temporarily kept: $\langle S^{\prime }\left(i,j\right)\rangle =\langle p_{\left[S_{1},S_{2},{\bar{S}}_{4},S_{5}\right]}\rangle$. This gives a set of two equations: (4)$$\begin{array}{ccc} \langle B^{\prime }\left(i,j\right)\rangle & = & 1-\langle \left(1-p_{1}B\left(i,j\right)\right)\left(1-p_{2}B\left(i,j+1\right)\right)\left(1-p_{4}^{\prime }R\left(i-1,j\right)\right)\left(1-p_{5}B\left(i-1,j\right)\right)\rangle , \end{array}$$(5)$$\begin{array}{ccc} \langle R^{\prime }\left(i,j\right)\rangle & = & 1-\langle \left(1-p_{1}R\left(i,j\right)\right)\left(1-p_{2}R\left(i+1,j\right)\right)\left(1-p_{4}^{\prime }B\left(i,j-1\right)\right)\left(1-p_{5}R\left(i,j-1\right)\right)\rangle . \end{array}$$
The approximation now enters the evaluation of the products on the right hand side of the equation. Each variable is replaced by its average and the spatial dependence is dropped: 〈B(i,j)〉↦〈B〉 and 〈R(i,j)〉↦〈R〉. Further, correlations are neglected so that the average of a product is just the product of averages. The expansions of (Equation 4) and (Equation 5) in powers of 〈B〉 and 〈R〉 are then readily obtained. Forgetting for a moment the intricacy linked to transversal splitting/collisions, the general expression for the dummy variables 〈S〉 and $\langle S^{\prime }\rangle$ reads:(6)$$\langle S^{\prime }\rangle =\sum_{k}p_{k}\langle S_{k}\rangle -\sum_{k_{1},k_{2}}p_{k_{1}}p_{k_{2}}\langle S_{k_{1}}\rangle \langle S_{k_{2}}\rangle +h.o.t.$$
with $p_{k}\in \left\{p_{1},p_{2},p_{4}^{\prime },p_{5}\right\}$ and where h.o.t. stands for the higher order terms, formally cubic, quartic, etc. The first sum in (Equation 6) corresponds to the contribution of the elementary configurations introduced in Figure 6, and the second sum to binary configurations, in particular the nontrivial ones corresponding to transversal splittings and collisions examined in Figure 8 (right). Orders of magnitude among the $p_{k}$, further support neglecting the contribution of configurations populated with three or more active sites, involving products of three or more probabilities $p_{k}$, and among contributions of a given degree, those not containing $p_{5}$ when compared to those that do, recalling the assumption $p_{5}≲1$ and $\{p_{1},p_{2}\}\ll 1$ implied by the nearly deterministic propagation of states in position 5 of Figure 6. A number of terms can, therefore, be neglected in the expanded forms of (Equation 4) and (Equation 5), which after simplification read: (7)$$\begin{array}{ccc} \langle B^{\prime }\rangle & = & \left(p_{1}+p_{2}+p_{5}\right)\langle B\rangle +p_{4}^{\prime }\langle R\rangle -p_{5}\left(p_{1}+p_{2}\right){\langle B\rangle }^{2}-{p_{5}}^{2}\langle B\rangle \langle R\rangle \;, \end{array}$$(8)$$\begin{array}{ccc} \langle R^{\prime }\rangle & = & \left(p_{1}+p_{2}+p_{5}\right)\langle R\rangle +p_{4}^{\prime }\langle B\rangle -p_{5}\left(p_{1}+p_{2}\right){\langle R\rangle }^{2}-{p_{5}}^{2}\langle R\rangle \langle B\rangle \;. \end{array}$$
This system presents itself as the discrete time counterpart of the differential system introduced in [14] to interpret the symmetry-breaking bifurcation observed at decreasing Re in the simulations. As a matter of fact, subtracting 〈B〉 and 〈R〉 on both sides of (Equation 7) and (Equation 8) respectively, one gets: (9)$$\begin{array}{ccc} \langle B^{\prime }\rangle -\langle B\rangle \approx \frac{d\langle B\rangle }{\left(dt\equiv 1\right)} & = & \left(p_{1}+p_{2}+p_{5}-1\right)\langle B\rangle +\cdots \end{array}$$(10)$$\begin{array}{ccc} \langle R^{\prime }\rangle -\langle R\rangle \approx \frac{d\langle R\rangle }{\left(dt\equiv 1\right)} & = & \left(p_{1}+p_{2}+p_{5}-1\right)\langle R\rangle +\cdots \end{array}$$
to be compared with system (1,2) in [14], reproduced here for convenience: (11)$$\begin{array}{c} \frac{dX_{+}}{dt}=aX_{+}+cX_{-}-bX_{+}^{2}-dX_{+}X_{-}\;, \end{array}$$(12)$$\begin{array}{c} \frac{dX_{-}}{dt}=aX_{-}+cX_{+}-bX_{-}^{2}-dX_{-}X_{+}\;, \end{array}$$
where $X_{\pm }$ represents what are now the densities 〈B〉 and 〈R〉. The coefficients in (Equation 11) and (Equation 12) are then related to the probabilities introduced in the model as $a\propto p_{1}+p_{2}+p_{5}-1$, $b\propto p_{5}\left(p_{1}+p_{2}\right)$, $c\propto p_{4}^{\prime }$, and $d\propto p_{5}^{2}$. By omitting the common proportionality constant that accounts for the time-stepping inherent in the discrete time reduction (featured by the denominator of left-hand sides in (Equation 9) and (Equation 10) as “(dt≡1)),” constants *a*, *b*, *c*, and *d* will serve as short-hand notation for the corresponding full expressions in terms of the probabilities $p_{k}$.

Since fixed points given by the condition $\langle S^{\prime }\rangle =\langle S\rangle$ is strictly equivalent to $dX_{\pm }/dt=0$, we can next take advantage of the analysis performed in [14] and predict a supercritical symmetry-breaking bifurcation for an order parameter |〈B〉−〈R〉| (denoted “*A*” in [14]) at a threshold given by $c^{\mathrm{cr}}=a\left(d-b\right)/\left(d+3b\right)$. This symmetry-breaking bifurcation takes place for $p_{4}^{\prime }=c>0$, but the model can deal with the regime below event A at Re≈800 for which $p_{4}^{\prime }\equiv 0$. In that case the bifurcation corresponding to global decay at Reg takes the form of two coupled equations generalizing (Equation 1) for DP. Using the abridged notation, these equations read:(13)$$\langle B^{\prime }\rangle =\left(a+1\right)\langle B\rangle -b{\langle B\rangle }^{2}-d\langle B\rangle \langle R\rangle \;,\;\langle R^{\prime }\rangle =\left(a+1\right)\langle R\rangle -b{\langle R\rangle }^{2}-d\langle R\rangle \langle B\rangle .$$
In addition to the trivial solution ${\langle R\rangle }_{0}={\langle B\rangle }_{0}=0$ corresponding to laminar flow, we have two kinds of non-trivial solutions, either single-sided (∗) with 〈R〉≠0 and 〈B〉=0 or 〈B〉≠0 and 〈R〉=0, the non-vanishing solution being ${\langle S\rangle }_{*}=a/b$, with S=R or *B*, or double-sided (∗∗) with ${\langle B\rangle }_{**}={\langle R\rangle }_{**}=a/\left(b+d\right)$. A straightforward stability analysis of the fixed points of iterations (Equation 13) shows that the one-sided solution is stable when b<d and unstable otherwise whereas the reversed situation holds for the two-sided solution. Returning to probabilities, the global stability threshold is thus given for a=0; hence, ${\left(p_{1}+p_{2}+p_{5}\right)}^{\mathrm{cr}}=1$ and the one-sided solution is expected when b<d; i.e., $p_{1}+p_{2}<p_{5}$. Results of the mean-field approach adapted from [14] to the present formulation will be illustrated in Figure 14 below.

### 2.4. Numerical Simulations

While serving as a guide to the exploration of a vast range of parameters, the simplified mean-field theory developed above is not expected to give realistic results relative to the critical properties expected near the transition point, whether decay at Reg or symmetry restoration above Re2. For example, observations suggest that LTB propagation is a dominant feature; hence, $p_{5}≲1$ and $\left\{p_{1},p_{2}\right\}$ is small, leading us to expect stable one-sided solutions systematically. This conclusion, however, strongly relies on neglecting all terms beyond second degree in (Equation 4) and (Equation 5) in the evaluation of the contribution of densely populated configurations, leading to (Equation 7) and (Equation 8). This is legitimate only when ${\langle S\rangle }^{n}\ll {\langle S\rangle }^{2}$, i.e., 〈S〉≪1, that is, close to decay in the case of a continuous (second-order) transition but not necessarily elsewhere in the parameter space, in particular at the one-sided/two-sided bifurcation where both 〈R〉 and 〈B〉 are of the same order of magnitude but may be large. Even when keeping the assumption of independence of contributions to the future state at a given lattice node, this problem is not easily addressed and, at any rate, has to be properly accounted for in the presence of stochastic fluctuations, which will be done numerically.

The translation of the probabilistic rules introduced in Section 2.2 using Matlab^®^ is straightforward once the “*B*/*R*/∅” convention is appropriately translated into “+1/−1/0”. No assumption is made other than the independence of the contributions of the different configurations to the outcome at a given lattice node, by strict application of the rules expressed through (Equation 2) and (Equation 3). In particular, computations involve the contribution of all configurations and not only the unary or binary ones, as presumed to derive the mean-field equations. Periodic boundary conditions have been applied to 2D lattices of various dimensions ($N_{B}\times N_{R}$), where $N_{B}$ ($N_{R}$) is the number of sites in the propagation direction of *B* (*R*) active states, with ordinarily $N_{B}=N_{R}$. At each simulation step, we shall measure the mean activity of *B* and *R* states denoted 〈B〉 and 〈R〉 above and from now on called turbulent fractions, as $F_{t}\left(B\right)={\left(N_{B}N_{R}\right)}^{-1}\#\left(B\right)$ and $F_{t}\left(R\right)={\left(N_{B}N_{R}\right)}^{-1}\#\left(R\right)$ where #(B) and #(R) are the numbers of sites in the corresponding active state.

A preliminary study of the model in a small domain has shown that the different transitional regimes and the symmetry-breaking bifurcation were indeed present as expected from the simplified mean-field approach. (We remind that the model contains nothing appropriate for organized laminar–turbulent regimes for Re>1200 and is relevant only for the strongly intermittent sparse LTB networks pictured in Figure 2 and Figure 3). In [14], we argued that the onset of transversal splitting was the source of genuinely 2D behavior. Accordingly we shall consider the stochastic model in two steps, below and above the onset of transversal splitting, here associated with $p_{4}^{\prime }\equiv 0$ and $p_{4}^{\prime }>0$ respectively. Furthermore, in the simulations the LTBs were seen to propagate obliquely with respect to the background downstream current. This propagation is nearly all contained in the probability attached to configuration $C_{5}$ ($p_{5}$ for propagation and $1-p_{5}$ for decay or slowing-down), and to a lesser extent influenced by the contribution of configuration $C_{1}$, mostly associated with in-line longitudinal splitting. We shall account for the limited sensitivity of the propagation speed to the value of Re to fix $p_{5}$ constant and close to 1, more specifically $p_{5}=0.9$, and let other parameters vary. The role of $p_{2}$ and $p_{4}^{\prime }$, both related to 2D features, will be studied separately in the two next sections.

## 3. Before Onset of Transversal Splitting, $P_{4}^{\prime }=0$


### 3.1. Coarsening from Two-Sided Initial Conditions

In the absence of transversal splitting, changes in the population of each state only comes from transversal collisions. As documented in [14], when starting from an initial condition with two similarly represented orientations, collisions lead to the formation of domains uniformly populated by one of each species, following from a majority rule, with interactions limited to the domain boundaries. A coarsening takes place with one species progressively disappearing to the benefit of the other, leaving a single-sided state at large times. The process is illustrated here using simulations of the model with $p_{5}=0.9$, $p_{1}=0.1$, $p_{2}=0.07$, values known from the preliminary study to produce a sustained nontrivial final state.

The decay from a fully active state populated with a random distribution of *B* and *R* states in equal proportions is scrutinized in a 256×256 domain with periodic boundary conditions. Figure 8 illustrates a particularly long transient displaying the different stages observed during a typical experiment.

The upper panel displays the time series of the turbulent fractions for each species, *B* and *R*, for a two-sided high-density initial condition, $F_{t}\left(B\right)+F_{t}\left(R\right)=1$, $F_{t}\left(B\right)≃F_{t}\left(R\right)≃0.5$. Contrasting with the monotonic variation observed when starting from one-sided initial conditions, either increasing from a low density of active states ($F_{t}=0.05$) or decreasing from a fully active configuration ($F_{t}=1$), the turbulent fractions change in a more complicated way that is easily understood when looking at the bottom line of snapshots. The total turbulent fraction first decreases due to the dominant effect of collisions. These collisions tend to favor a spatial modulation of the activity amplifying inhomogeneities in the initial conditions. This distribution results from the majority effect expressing the local stability of one-sided states predicted by the mean-field analysis. A periodic pattern already appears at t=100, with bands oriented parallel to the second diagonal of the square domain. *B* states move right along the horizontal axis, and *R* states up along the vertical axis, at the same average speed so that the pattern drifts along the first diagonal of the domain. Regions where *B* or *R* dominate are locally stable against destructive collisions and activity is limited to B/R interfaces. After a while, splittings begin to counteract collisions and an overall activity recovers, here for t≈250. The local density of *B* and *R* states increases inside bands that become better defined, reaching a sustained regime with two *R*–*B* alternations, wide and narrow, at t≃1500. This configuration is nearly stable and slowly evolves only due to the erosion of narrowest bands at the R/B interfaces. At t≈5500 these bands disappear by merging, leaving two bands, *B* wide and *R* narrow. The same slow erosion process leads to the final homogeneous *B* regime by decay of the *R* band at t∼96,000. The two successive band decays take place at roughly constant total turbulent fraction with fast adjustment at the band decay, up to the final single-sided turbulent fraction. The asymptotic state is independent of the way it has been obtained, from one-sided or two-sided initial conditions.

The long duration of the transient taken as an example is due to the near stability of the rather regular pattern building up after the initial fast decay. This property is in fact the result of a geometrical peculiarity of the square domain: *B* and *R* states travel statistically at the same speed through the domain, horizontally and vertically, respectively, so that the band integrity is maintained despite propagation and the evolution controlled by collisions at the *B*–*R* and *R*–*B* interfaces only. The observed slow erosion process only results from large deviations among collisions. In rectangular domains, the propagation times become different and the symmetry of the two interfaces is lost. A bias results, which induces a systematic erosion of bands and a shorter transient duration. Whatever the aspect ratio, one of the states is always ultimately eliminated and the last stage of the transient corresponds to a trend toward a statistically uniform saturated one-sided regime with a turbulent fraction strictly independent of the shape. Accordingly, to save the time corresponding to the transient, in the next section we will study the decay of the one-sided regime by starting from random one-sided initial conditions.

All these features nicely fit the empirical observations discussed at length in [14] where similar transients were obtained below the onset of transversal splitting—in much smaller effective domains and with far fewer interacting LTBs, however (Figure 2, right panel, and Figure 4, left panels).

### 3.2. Decay: 1D vs. 2D

The model is designed to exemplify a decay according to the DP scenario in a two-dimensional setting, with specificities linked to the anisotropic propagation properties of the LTBs in transitional channel flow, and, in particular, propose an interpretation for the observation of 1D-DP exponents in the absence of transversal splitting ($p_{4}^{\prime }=0$). Accordingly, we examine the role of transverse diffusion (parameter $p_{2}$) modeling the slight upstream shift that may affect LTBs at longitudinal splitting. We focus on a set of experiments with $p_{5}=0.9$, $p_{2}$ fixed, and control parameter $p_{1}$. When $p_{2}$ cancels exactly, it is easily understood that transversal expansion is forbidden: An active *B* state at (i,j+1) or *R* state at (i+1,j) at time *t* cannot give birth to an active state of the same kind at (i,j) at t+1. The evolution stems from processes associated with configuration $C_{5}$ with probability $p_{5}$ or $C_{1}$ with probability $p_{1}$. These processes change occupancy only along direction *i* for *B* states, and *j* for *R* states, precisely in the direction corresponding to the single-sided regime considered (after termination of the transient). The dynamics are, therefore, strictly one-dimensional and decay is expected to follow the 1D-DP scenario. In contrast, introducing some transverse diffusion ($p_{2}\neq 0$) immediately gives some 2D character to the dynamics. This is illustrated in Figure 9, Figure 10, Figure 11 and Figure 12.

We consider first $p_{2}$ non-zero and relatively large $p_{2}=0.1$. Figure 9 displays the behavior of the turbulent fraction as a function of $p_{1}$.

The left panel illustrates the decrease of the turbulent fraction with the number of steps from a uniformly fully turbulent single-sided state ($F_{t}=1$ at t=0) in a domain D=(192×192), showing the saturation to a finite value $\langle F_{t}\rangle$ above threshold, a near power-law decay close to threshold, and an exponential decay below. The right panel presents the mean of $F_{t}$ after elimination of an appropriate transient as a function of $p_{1}$, for simulations in domains up to 512×512 for the lowest values of $F_{t}$. Once fitted in the range $p_{1}\in \left[0.058,0.064\right]$ against the expected power law behavior $\langle F_{t}\rangle =a{\left(p_{1}-p_{1}^{c}\right)}^{\beta }$ one gets a=3.213(2.936,3.489), $p_{1}^{c}=0.05844$(0.05842,0.05845), β=0.5811(0.566,0.5962), in very good agreement with the value $\beta_{\mathrm{DP}}\approx 0.584$ when D=2 [2]. This is confirmed in the inset of Figure 9 (right) showing ${\langle F_{t}\rangle }^{1/0.584}$ as a function of $p_{2}$ for $F_{t}$ small, the linear variation of which extrapolates to zero for $p_{1}\approx 0.05843$.

Having a good estimate of the threshold one can next consider the decay of the turbulent fraction, which is supposed to decrease as a power law at criticality, $p_{1}=p_{1}^{c}$: $F_{t}∼t^{-\alpha_{\mathrm{DP}}}$ with $\alpha_{\mathrm{DP}}=\beta_{\mathrm{DP}}/{\nu_{\|}}_{\mathrm{DP}}$ where ${\nu_{\|}}_{\mathrm{DP}}\approx 1.295$; hence, $\alpha_{\mathrm{DP}}\approx 0.451$ [2]. Figure 10 (left) shows that this is indeed the case for the compensated turbulent fraction $F_{t}\times t^{\alpha_{\mathrm{DP}}}$, up to the moment when fluctuations become too important due to size effects and lack of statistics. When $p_{1}$ is different from $p_{1}^{c}$ but stays sufficiently close to it, the variation of the turbulent fraction keeps trace of the critical situation, except that the number of steps needs to be rescaled by the distance to threshold due to critical slowing down: the time scale τ diverging as ${\left(p_{1}-p_{1}^{c}\right)}^{-\nu_{\|}}$, number of steps is rescaled upon multiplying it by ${\left(p_{1}-p_{1}^{c}\right)}^{\nu_{\|}}$. Figure 10 (right) indeed shows a good collapse of the compensated curves as a function of the rescaled number of steps when using the exponents corresponding to 2D-DP, $\alpha_{\mathrm{DP}}\approx 0.451$ and ${\nu_{\|}}_{\mathrm{DP}}\approx 1.295$ [2].

We now consider $p_{2}=0$ which, as argued earlier, should fit the critical behavior of directed percolation when D=1. In that case, when using square or nearly-square rectangular domains, size effects turn out to be particularly embarrassing as will be illustrated quantitatively soon. However, we can take advantage of the fact that, assuming propagation in the one-sided regime, e.g., along the direction for *B* active states, $N_{B}$ being the corresponding number of sites involved, the computed turbulent fraction is, in fact, the average of the activity over $N_{R}$ independent lines in the complementary direction, while still being sensitive to size effects controlled by $N_{B}$. Accordingly, at given computational load (proportional to $N_{B}\times N_{R}$), one can freely increase the size artificially in considering a strongly elongated domain $D^{\prime }=\left[\left(N_{B}\times k\right)\times \left(N_{R}/k\right)\right]$, with *k* sufficiently large that the average over $N_{R}/k$ independent lines still make sense from a statistical point of view, while postponing size effects. With reference to a (192×192) domain, we have obtained good results with k=16, i.e., 3072×12 up to k=64, i.e., 12288×3.

Though this choice is a bit extreme, we present here results about 1D-DP criticality with the 12288×3 domain in Figure 11. The left panel displays the variation of the mean turbulent fraction with $p_{1}$, which has been fitted against the expected power law, $\langle F_{t}\rangle =a{\left(p_{1}-p_{1}^{c}\right)}^{\beta }$. One gets a=1.473(1.446,1.5), $p_{1}^{c}=0.2682$(0.2682,0.2683), β=0.2701(0.2664,0.2738). This value of β is quite compatible with the value $\beta_{\mathrm{DP}}\approx 0.276$ when D=1 [2]. Furthermore, accepting this value, a linear fit of ${\langle F_{t}\rangle }^{1/\beta }$ with $p_{1}$ then provides an extrapolated threshold $p_{1}^{c}=0.26817$. As seen in the right panel of Figure 11, in the neighborhood of $p_{1}^{c}$ a good collapse is obtained for the compensated turbulent fraction as a function of the rescaled number of steps when using the exponents α=0.159 and $\nu_{\|}=1.734$ corresponding to 1D-DP [2].

Size effects already alluded to above are illustrated in Figure 12.

Displaying the turbulent fraction as a function of the number of steps for linear size $N_{B}$ from small systems to relatively large ones ($N_{B}=64$ up to 768) in lin-log scale, the left panel illustrates the late stage of decay right at criticality as obtained from the previous study summarized in Figure 11. It is seen that, in the time-window considered $\left(0,10^{5}\right)$ the exponential dependence observed at small sizes is progressively replaced by the power-law behavior expected at criticality at infinite size. Size effects are also ruled by scaling theory; see, e.g., [2] for DP. They relate to correlations in physical space that are associated with exponent $\nu_{⊥}$. The ratio $z=\nu_{\|}/\nu_{⊥}$ is called the dynamical exponent and theory predicts that, for finite size systems, scaling functions depend on time with the number of sites as $t^{D/z}/N$ where *N* is the total number of sites. In the (quasi-)one-dimensional regime we are interested in, D=1, *N* is just $N_{B}$ and z=1.58 [2]. The right panel of Figure 12 indeed shows extremely good collapse of the traces corresponding to those in the left panel, once the number of steps is rescaled as $t/N_{B}^{1.58}$ and the turbulent fraction is compensated for decay as $F_{t}\left(t\right)\times t^{0.159}$, both exponents taking on the 1D-DP values already mentioned.

Of interest in the context of channel flow decay, the crossover from 2D behavior for $p_{2}$ sizable (e.g., $p_{2}=0.1$, Figure 9 and Figure 10) to 1D behavior for $p_{2}=0$ is of interest since $p_{2}$ is associated with the progressive importance of off-aligned longitudinal splitting as Re increases. A series of values of $p_{2}$, decreasing to zero roughly exponentially, has been considered and the corresponding DP threshold has been determined as given in Table 1 and shown in Figure 13 (left).

Except for $p_{2}=0$ determined as explained above (Figure 11), these values have been obtained in domains 192×192 with averaging over 10 independent experiments. Figure 13 (right) displays the averaged time-series of the turbulent fraction at criticality for each of these values of $p_{2}$, once compensated for decay according to 2D-DP (α=0.451).

The results for $p_{2}=0$, evolving as $t^{\alpha_{2D}-\alpha_{1D}}$ are marked with (∗) and (∗∗) are obtained in the 192×192 domain and in the 3072×12 quasi-1D domain, respectively. In the time span considered here, the latter is free from finite-size effects which is not the case of the former with the corresponding compensated data decaying exponentially at the largest times. It is easily seen that, except for $p_{2}=0$, the compensated time series display a wide plateau indicating that 2D behavior holds for a certain amount of time. Whereas traces for $p_{2}=0.1$ and $p_{2}=0.05$ cannot be distinguished, for smaller values of $p_{2}$ the plateau regime starts at larger and larger times and develops after having followed the 1D trace for longer and longer durations, clearly indicating the influence of the anisotropy controlling the effective dimensional reduction. A similar consequence of the crossover affects the decrease of the mean turbulent fraction with the distance to threshold but, apart from this qualitative observation, no reliable information can be obtained on exponent β owing to the difficulty to reach the relevant critical regime.

We shall not document the case when $p_{1}=0$ and $p_{2}$ varies. This situation is not observed in the simulations since off-aligned longitudinal splitting is conspicuous only sufficiently above Reg, in the vicinity of which decay is fully accounted for by in-line longitudinal splitting modeled by a variable $p_{1}\neq 0$, but the possibility remains, at least conceptually. The decay when $p_{1}=0$ happens to follow the same 1D-DP scenario though the argument is slightly less immediate. It relies on the observation that no growth is possible in the propagation direction of a given LTB species, whereas off-aligned longitudinal splitting ($p_{2}\neq 0$) permits growth and diffusion in the transverse direction. Under the combined effects of transversal diffusion ($p_{2}$ small) and propagation ($p_{5}$ large), near-threshold, the sustained turbulent regime is made of quasi-1D clusters that are aligned with and drift along the diagonal of the lattice, i.e., the stream-wise direction, and get thinner and thinner when decaying, supporting the reduction to a “D=1” scenario. Here, the trick used for $p_{2}=0$ does not work, and simulations in square domains are necessary with no escape for size effects which hinders the observation of the critical regime. Nevertheless, $p_{2}^{c}$ when $p_{1}=0$ seems close to $p_{1}^{c}$ when $p_{2}=0$, suggesting some symmetry between $p_{1}$ and $p_{2}$.

The relevance of the results with $p_{4}^{\prime }\equiv 0$ to transitional channel flow will be discussed in the concluding section. We now turn to the general two-sided case with transversal collisions and splittings.

## 4. Beyond Onset of Transversal Splitting, $P_{4}^{\prime }>0$


In statistical thermodynamics systems, critical properties at a second order phase transition leads to define a full set of exponents governing the variation of macroscopic observables close to criticality [26]. The concept of universality was introduced to support the observation that these systems can be classified according to the value of their exponents depending on a few qualitative characteristics, the most prominent ones being the symmetries of the order parameter and the dimension of physical space. This viewpoint can be extended to far-from-equilibrium systems such as coupled map lattices (CMLs) displaying nontrivial collective behavior. The associated ordering properties present many characteristics of thermodynamical critical phenomena at equilibrium. Universality classes beyond those known from equilibrium thermodynamics have been shown to exist with different sets of exponents. An additional criterion, the synchronous or asynchronous nature of the dynamics, has been found relevant to distinguish among them [27]. In the context of the present model, as soon as probability $p_{4}^{\prime }$ grows from zero, fully one-sided configurations previously reached after the termination of a possibly long transient are now unstable against the presence of states with the complementary color. The stationary regime that develops in the long term can be, either ordered, i.e., one-sided with one dominant active state (*B* or *R*), or disordered, i.e., two-sided with statistically equal fractions of each active state (*B* and *R*). Furthermore, a transition at some critical value ${p_{4}^{\prime }}^{c}$ is expected to take place on general grounds. This gives us the motivation to study the response of the model to the variation of $p_{4}^{\prime }$ as a critical phenomenon beyond the mean-field expectations of Section 2.3.

The results of the mean-field approach, system (Equation 11) and (Equation 12), rephrased from [14], are depicted in Figure 14 (left). Upon variation of parameter *c* representing $p_{4}^{\prime }$ up to an unknown rescaling factor, all along the one-sided regime ($c<c^{\mathrm{cr}}$), the total turbulent fraction is seen to decrease while the order parameter measuring the lack of symmetry similarly decreases to zero according to the usual Landau square-root law. Obviously symmetrical, the two-sided regime ($c>c^{\mathrm{cr}}$) is then characterized by a total turbulent fraction that regularly grows due to the contribution of splitting, whatever the type of active state.

From now on, we shall simply refer to the turbulent fractions and other statistical quantities as their time average over a sufficiently long duration, up to $2\times 10^{6}$ simulation steps, after elimination of an appropriate transient, up to $10^{5}$ steps, the largest values being necessary close to the transition point owing to the well-known critical slowing down. On the one hand, the total turbulent fraction is obviously defined as $\bar{F_{t}\left(B\right)+F_{t}\left(R\right)}$, where the over-bar denotes the time averaging operation. (Later on, we shall omit this over-bar when no ambiguity arises between the instantaneous value of a quantity and its time average, especially for the axis labelling in figures.) On the other hand, the lack of symmetry can be measured by the signed difference averaged over time $\bar{F_{t}\left(B\right)-F_{t}\left(R\right)}$, able to distinguish global *B* orientation from its *R* counterpart, or rather its absolute value $\left|\bar{F_{t}\left(B\right)-F_{t}\left(R\right)}\right|$ since we are only interested in the amplitude of the asymmetry (called ‘*A*’ in [14]) and not in which orientation is dominant, the two being equivalent a priori for symmetry reasons. However, due to the finite size of the system, in the symmetry-broken regime close to threshold, orientation reversals can be observed as illustrated later (Figure 15), so that blind statistics in the very long durations are no longer representative of the actual ordering. Like in thermal systems [28] or their non-equilibrium counterparts [27], it is thus preferable to define the order parameter through the mean of the unsigned difference: $\bar{\;|F_{t}\left(B\right)-F_{t}\left(R\right)|\;}$. Corresponding simulation results are displayed in Figure 14 (right) for a system of size (256×256). The general agreement between the two diagrams is remarkable, up to an unknown multiplicative factor translating *c* into $p_{4}^{\prime }$, as discussed earlier. One can notice that the order parameter is minimal but not zero in the two-sided regime, which is due to fluctuations and the fact that the two operations of averaging over time and taking the absolute value do not commute. Finite-size effects are also apparent as a rounding of the graph at the location of the would-be critical point in the thermodynamic limit.

The current justification for taking the absolute value is that the time between orientation reversals diverges with the system size and the phase transition only takes place once we have taken the thermodynamic limit of infinitely large systems studied over asymptotically long durations [28]. Accordingly, very long well-oriented intermissions can be considered as representative of the symmetry-broken regime. The problem is illustrated in Figure 15 displaying the time series of $F_{t}\left(B\right)-F_{t}\left(R\right)$ and histograms of $|F_{t}\left(B\right)-F_{t}\left(R\right)|$ for $p_{4}^{\prime }=0.0121$, still in the one-sided but already alternating regime, next for $p_{4}^{\prime }=0.0125$ and 0.0126, where one can notice a change in the shape of the histogram, and finally for $p_{4}^{\prime }=0.0140$, sufficiently deep inside the two-sided regime where the histogram displays a sharp maximum at the origin. On this basis one could use the histograms of the “order parameter” and determine the threshold from the position of its most probable value, whether non-zero in the symmetry-broken state or at the origin when symmetry is restored. This procedure would give ${p_{4}^{\prime }}^{c}\approx 0.01255$.

The symmetry-breaking bifurcation can now be studied beyond the mean-field description as other collective phenomena studied in equilibrium and far-from-equilibrium statistical physics: In addition to the order parameter, the variation of which leads to the definition exponent β in the ordered regime, another observable of interest is the susceptibility measuring the response to an applied field conjugate to the order parameter, vis. M=χH with the magnetization *M* coupled to magnetic field *H* in the case of magnets. The susceptibility diverges near the critical point, with leads to the definition of two exponents γ and $\gamma^{\prime }$ in the disordered and ordered regime, respectively. Universality implies $\gamma =\gamma^{\prime }$, as can already easily be derived in the mean-field framework. When a conjugate field cannot be defined, one uses the property that fluctuations take the instantaneous value of the order parameter away from its average value, which can be understood as resulting from the response to a conjugate field. This helps one to relate the susceptibility to the variance of fluctuations of the order parameter. The identification is up to a multiplication by the “volume” of the system that has to be introduced in order to compare the results from systems with different sizes. This is what will be done here; hence, $\chi =N_{B}N_{R}\times \mathrm{var}\left(|F_{t}\left(B\right)-F_{t}\left(R\right)|\right)$. As shown in Figure 16 (top), this quantity displays a sharp maximum, indicative of the singularity expected at the thermodynamic limit. In a finite-size but large system, the critical point is then estimated from the position of the maximum of the susceptibility. Here, this gives ${p_{4}^{\prime }}^{c}\approx 0.0123$ slightly smaller but compatible with the value obtained above from the examination of the histograms. Unfortunately, this discrepancy due to size-effects forbids us to determine exponents β and γ with some confidence.

Having in mind results of the mean-field approach, namely, β=1/2 and γ=1, we can, however, estimate the range where stochastic fluctuations have nontrivial effects. The bottom-left panel of Figure 16 displays the variation of the order parameter already shown in Figure 14 (right), but now squared in order to show that, far from the critical point, the system fulfils the mean-field square-root prediction to an excellent approximation, with an extrapolated threshold ${p_{4}^{\prime }}^{\mathrm{MF}}≃0.0133$, shifted upwards with respect to the estimates obtained from the simulations ${p_{4}^{\prime }}^{c.}≃0.0123$–0.0125.

In the same way, the divergence of the susceptibility with exponents $\gamma =\gamma^{\prime }=1$ expected from the mean-field argument shows up upon retreating the data already given in Figure 16 (top) and plotting 1/χ as a function of $p_{4}^{\prime }$. This is done in Figure 16 (bottom-right) showing the same linear variation of 1/χ below and above the transition point, in agreement with the theory. The extrapolation of the linear fits on both sides of the transition yield ${p_{4}^{\prime }}^{c}\approx 0.0127$ in reasonable agreement with the value obtained from the order parameter variation in the same conditions and definitely larger than the empirical values. Clearly, deviations seen in the boxed parts of these two figures warrant further scrutiny, motivating our current approach via finite-size scaling theory [28] in search for universality. On going work attempts at a full characterization of the critical regime through exponents determination. Though local agents do not behave as Ising spins, symmetries are basically identical, so that the equilibrium 2D Ising universality class or its non-equilibrium extension [27] might be relevant. We shall discuss this further below.

## 5. Discussion and Concluding Remarks

Coming long after a conjecture by Pomeau [1], empirical evidence is growing that the ultimate stage of decay of wall-bounded turbulent flows towards the laminar regime follows a directed-percolation scenario. The evidence comes from laboratory experiments and direct numerical simulation of the Navier–Stokes equations but this support is still far from a theoretical justification. The recognition of the globally subcritical character of nontrivial states away from laminar flow and the elucidation of the structure of coherent structures involved in these nontrivial states [29] were first steps in this direction. The next ones would be the elucidation of special phase space trajectories from sustained localized turbulence accounting for the decay to laminar regime, on one side, and to proliferation via splitting, on the other side, using specific algorithms for the detection of rare events and the determination of transition rates that can be attached to them (see [30] for an illustrative example and references). These are heavy, and possibly not much rewarding, tasks but it would be nice to be able to attach numbers to specific events such as the splittings illustrated in Figure 3 or Figure 4. We have chosen to short-circuit such studies through analogical modeling, by which seemed more appropriate to make further progress regarding the thermodynamic limit and associated universality issues. One should though consider this practice as providing hints and not a demonstration that the results will apply to the case under study.

In the present paper, the problem has been considered from this last viewpoint, assuming that the ultimate decay stages were amenable to the most abstract level of implementation in terms of probabilistic cellular automata [2], following [20,22]. We focussed on the specific case of channel flow that offers a particularly rich transitional range. Its upper part displays regular non-intermittent laminar–turbulent patterns that can better be described using the tools of pattern-forming theory [15,25,31]. The lower transitional range is characterized by their spatiotemporally intermittent disaggregation, to which the considered type of modeling is particularly relevant. The analogy alluded to above has, however, been severely constrained to fit the empirical observations. The main assumptions were the introduction of two types of active agents attached to each kind of localized turbulent bands propagating in one of the two possible orientations with respect to the stream-wise direction. Interactions were assumed local so that the probabilistic cellular automata evolved simple nearest neighbors von Neumann neighborhoods (Figure 5, Figure 6 and Figure 7). Scrutiny of simulation results lead to the introduction of a certain number of probabilities governing the fate of single-occupancy neighborhoods. Multiple-occupancy was treated as a combination of single-occupancy configurations supposedly independent, reducing the number of parameters to be introduced and drastically simplifying the interactions (at any rate impractical to estimate in detail). A clear-cut physical interpretation was, however, given to each parameter in the set reduced to four, accounting for every possible stochastic event affecting the agents, namely, propagation, decay, and splitting, either longitudinal or transversal. A mean-field study of the model, neglecting the nontrivial effects of stochastic fluctuations, reproduced the empirical bifurcation diagram of channel flow at a qualitative level (Figure 14). Transitions have been studied quantitatively by numerical simulation of the stochastic model considering variations of these parameters as putative functions of the Reynolds number Re, highlighting three situations:

In the two first cases, the parameter $p_{4}^{\prime }$ associated with transversal splitting, i.e., the nucleation of a daughter with orientation opposite of its mother, was switched off, as inferred from observations for Re≲800, where the single-sided regime is well established. The coarsening observed when starting from two-sided initial conditions was faithfully reproduced (Figure 8) and decay seen to follow the directed-percolation expectations. The specific conclusion was that, when parameter $p_{2}$ is no-zero, with $p_{2}$ attached to longitudinal but upstream-shifted splitting, the scenario is typical of a 2D system with a high level of confidence, whereas when it is strictly zero, i.e., the daughter strictly aligned with the mother, the decay is 1D. A cross-over is observed when $p_{2}$ is reduced, that manifests itself as a transient reminiscent of 1D behavior, the longest the closest $p_{2}$ is to zero. Simulations of channel flow have shown that exponent β controlling the ultimate decay of the turbulent fraction was that of 1D directed percolation [16]. Since parameter $p_{2}$ is attached to the slight upstream trajectory shift experienced by a daughter upon splitting from its mother, this observation strongly suggests that the trajectory shift is mostly irrelevant and that localized turbulent bands propagate along independent tracks so that the end result is just a mean over the direction complementary to their propagation direction.

The last situation we have considered corresponds to $p_{4}^{\prime }\neq 0$, with transversal splitting on. This parameter measures the frequency of transversal splitting and is expected to increase with Re. Accordingly, the system can change from one-sided when $p_{4}^{\prime }$ is zero or small, to two-sided when it is large. The transition has indeed been observed and mean-field predictions were well observed far from the transition point. Unfortunately, while the effect of fluctuations close to that point was obvious, strong size effects have forbidden us to approach it and evaluate critical corrections. This is the subject of on-going work within the framework of finite-size scaling theory [2,27,28]. This follow-up should allow us to establish the universality class to which this transition belongs. Here, the left-right symmetry of localized turbulent bands with respect to the stream-wise direction is reminiscent of the up-down symmetry of magnetic systems at thermodynamic equilibrium, which may lead to conjecture the relevance of the 2D Ising class [26]. This class appears also applicable to coupled map lattices with the same up-down symmetry when updated asynchronously, one site after the other, close to randomization by thermal fluctuations. In contrast, another universality class is obtained with synchronous update [27]. Here, the situation is unclear: on the one hand, configurations are treated as a whole in a simulation step, which tips the scales in favor of a synchronous update model (in line with what is expected for a problem primitively formulated in terms of partial differential equations); on the other hand, spatial correlations generated by the deterministic dynamics governing the coupled map lattices are weakened by the independence of random drawings at the local scale, which can be viewed as a source of asynchrony in the probabilistic cellular automata. In its application to the symmetry-breaking bifurcation in channel flow, this uncertainty is, however, only of conceptual importance in view of size effects: owing to the large and unknown time-scale rescaling that allowed us to pass from flow structures to local agents in the model and to the narrowness of the region where critical corrections are expected, the mean-field interpretation developed in [14] appears amply sufficient.

In the three cases that were considered in detail (specific cuts in the parameter space), the transitions remained continuous. However, this may not always be the case since there are known example of similar systems displaying transitions akin to first-order ones [24]. Even while keeping the same general frame, a plethora of circumstances of physical interest can be mimicked: propagation can be made more stochastic by decreasing $p_{5}$, splitting rules not observed in channel flow can be considered, e.g., with $p_{3}$ or $p_{4}$ different from zero, etc., though it seems hard to anticipate situations where the universal features pointed out here would not hold. In contrast, when dealing with highly populated configurations, even in the simple nearest-neighbor von Neumann setting, rules can be made more complicated by introducing the neighborhood’s degree of occupation. This introduction might help us to account also for the upper part of the transitional range of wall-bounded flows characterized by the emergence of regular patterns in the same stochastic framework [11]. The construction of the present model is, of course, fully adapted to the study of universality in the framework of the theory of critical phenomena in statistical physics, especially directed percolation. Still, we are confident that the kind of approach illustrated here brings a valuable contribution to the understanding of the transition to turbulence, by rationalizing its key ingredients in an easily accessible way.

## Figures and Tables

**Figure 1 entropy-22-01348-f001:**
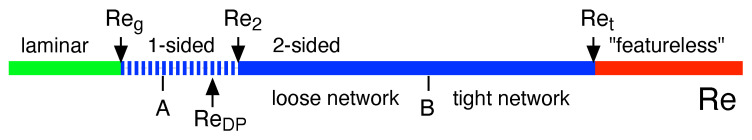
Bifurcation diagram of plane channel flow after [14]. Reg≈700. Transversal splitting sets in at Re∼ 800 (event A). The extrapolated 2D-DP threshold is ReDP≃984. The “one-sided → two-sided” transition takes place at Re2≃1011. localized turbulent bands (LTBs) exist up to Re≈1200 (event B), beyond which a continuous laminar–turbulent oblique pattern prevails up to the threshold for featureless turbulence Ret≈3900.

**Figure 2 entropy-22-01348-f002:**
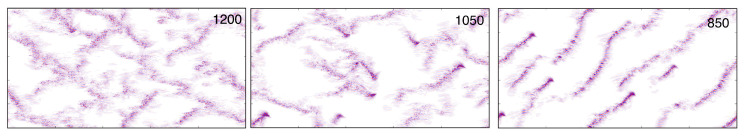
Illustration of the different regimes featuring the wall-normal velocity component at the mid-gap; turbulent/laminar flow is pink/white, after data in Figure 1 of [14]. The domain size is 250×500 (span-wise × stream-wise). The flow is from left to right. **Left**: Strongly intermittent loose continuous LTB network at Re=1200 (∼event B). **Centre**: Two-sided regime at Re=1050 (Re≳Re2). **Right**: One-sided regime at Re=850. Downstream active heads (DAHs) are easily identified in the two right-most panels; a single one is visible in the upper left corner of the left image, marking the transition between sustained regular patterns and loose intermittent ones. Images here and in Figure 2 and Figure 3 are adapted from snapshots taken out of the supplementary material of reference [14].

**Figure 3 entropy-22-01348-f003:**
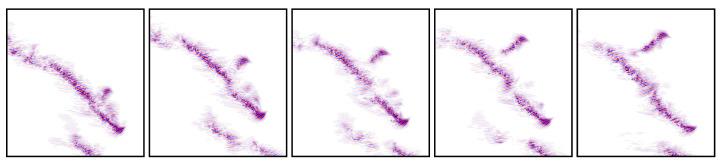
First observed occurrence of transversal splitting during a simulation at Re=800 for t∈ (17100:100:17500). The stream-wise direction is horizontal and the flow is from left to right.

**Figure 4 entropy-22-01348-f004:**
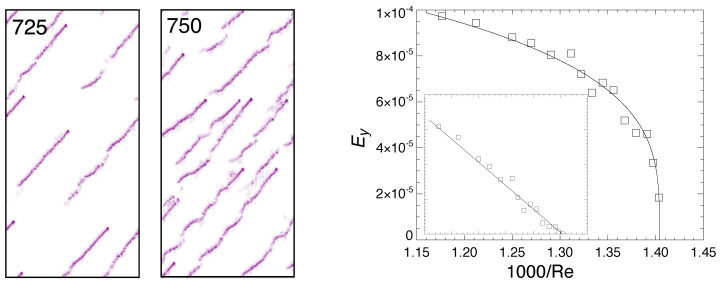
(**Left**): One-sided flow at Re=725, 750; same representation as in Figure 2. The domain size is now 500×1000, the stream-wise direction is vertical, and the flow upwards. (**Right**): Used as a proxy for the turbulent fraction, $E_{y}=V^{-1}\int u_{y}^{2}\;dV$ is displayed as a function of 1/Re; inset: same data raised at power 1/β with β=0.28 suggesting decay according to the DP scenario in 1D, adapted from [16].

**Figure 5 entropy-22-01348-f005:**
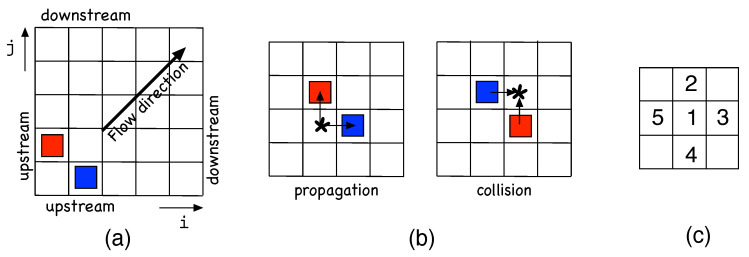
(**a**) Cellular automata lattice with the two types of active states, *B* and *R*; the state at an empty node is denoted ∅ and left blank. (**b**) Left: the two possible kinds of propagation from an initial position marked with the “∗”. Right: collision configuration to the point marked with the “∗”. (**c**) Labeling of the von Neumann neighborhood used to account for the dynamics.

**Figure 6 entropy-22-01348-f006:**
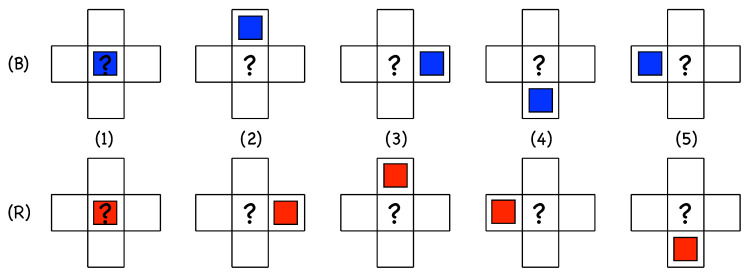
Single-color configurations: from the overall geometry depicted in Figure 5a, the downstream side of a state is to the top for *B* states and to the right for *R* states. Each colored square indicates the active state in the configuration at time *t* of site (i,j) at the center. The question mark features the probabilistic outcome (time t+1).

**Figure 7 entropy-22-01348-f007:**
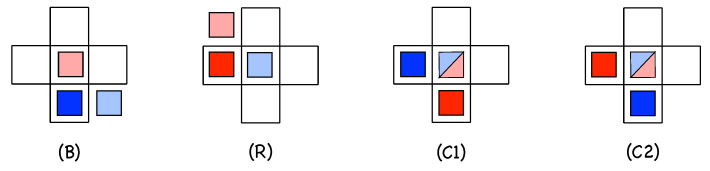
Modeling of transversal splitting for states of type (B) propagating horizontally and (R) propagating vertically, the base flow being along the diagonal (↗). Heavy colors indicate states present at time *t* and, playing the role attributed to question marks in Figure 6; light colors stand for states possibly present at time t+1 according to probabilities $p_{5}$ (propagation) and $p_{4}^{\prime }$ (transversal splitting). Conflicting configurations are (C1) ([SSSRB] corresponding to propagation leading to a collision and (C2) [SSSBR] corresponding to simultaneous transversal splittings, respectively (here S=∅ for clarity).

**Figure 8 entropy-22-01348-f008:**
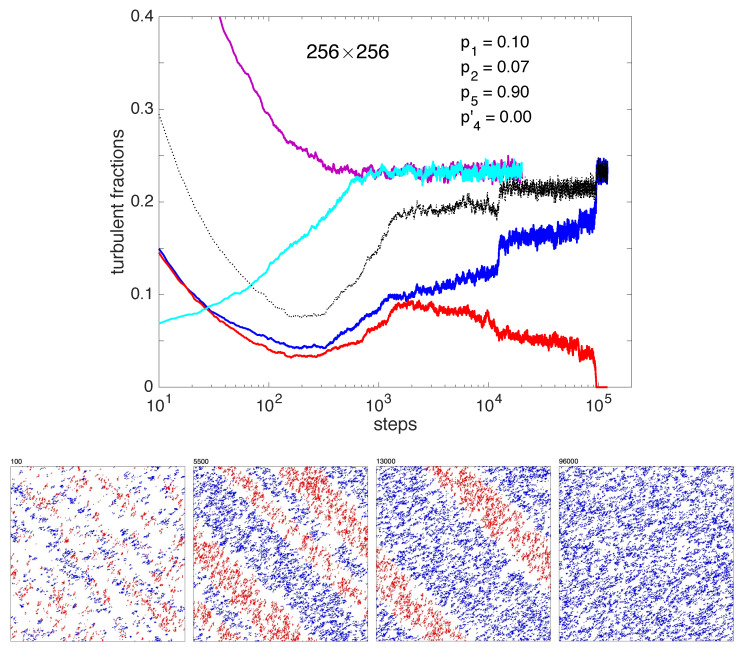
(**Top**): Time series of the turbulent fractions for a simulation from a fully active initial configuration with *B* and *R* states in equal proportions—blue and red in graphs, respectively; the dotted black trace is for the total turbulent fraction. Two simulations starting from low ($F_{t}=0.05$, cyan) and high ($F_{t}=1$, magenta) density one-sided states are displayed for comparison. (**Bottom**): Snapshot of state during the simulations from the two-sided initial condition, at t=100 during initial decay, at t=5500 with two pairs of active bands of each color, at t=13,000 when the narrowest bands merge and disappear, at t=96,000 when the *R* active band disappears, leaving a uniform *B* state.

**Figure 9 entropy-22-01348-f009:**
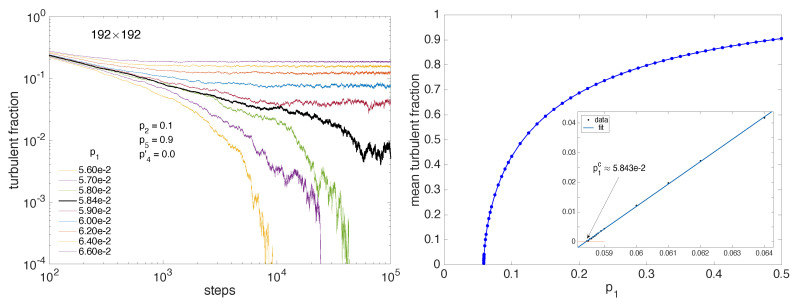
(**Left**): Time series of the turbulent fraction at different values of $p_{1}$; average over 5 (10) independent simulations ($p_{1}=0.0584\approx p_{1}^{c}$, black trace). (**Right**): Mean value of the turbulent fraciton at stationary state as a function of $p_{1}$ (original data). Inset: once raised to power 1/β, with $\beta =0.584\approx \beta_{\mathrm{DP}}$ for D=2, the mean turbulent fraction tends to 0 linearly with an extrapolated threshold $p_{1}^{c}=0.05843$.

**Figure 10 entropy-22-01348-f010:**
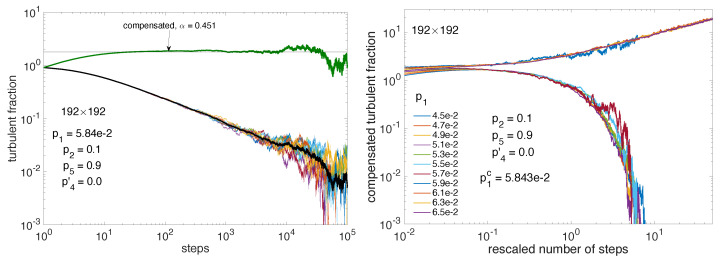
(**Left**): Power law decay of the turbulent fraction at $p_{1}=0.0584\approx p_{1}^{c}=0.05843$: compensation with $\alpha_{\mathrm{DP}}$ confirms the 2D nature of the process. (**Right**): Critical behavior near threshold: compensated turbulent fraction $\langle F_{t}\rangle \times t^{\alpha_{\mathrm{DP}}}$ as a function of the number of steps rescaled by ${\left(p_{1}-p_{1}^{c}\right)}^{{\nu_{\|}}_{\mathrm{DP}}}$ for $p_{1}\in$ [4.5:0.2:6.5] $\times 10^{-2}$ surrounding the presumed critical value $p_{1}^{c}$, with the exponents corresponding to DP for D=2.

**Figure 11 entropy-22-01348-f011:**
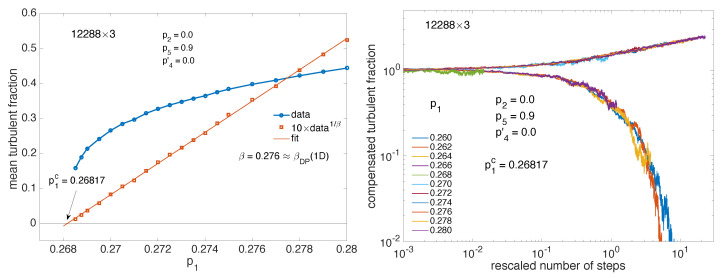
(**Left**): Mean turbulent fraction at stationary state as a function of $p_{1}$. Once raised at power 1/β with $\beta =0.276\approx \beta_{\mathrm{DP}}$ for D=1, the mean turbulent fraction tends to 0 linearly with an extrapolated threshold $p_{1}^{c}=0.26817$. (**Right**): Critical behavior near threshold: compensated turbulent fraction $\langle F_{t}\rangle \times t^{\alpha_{\mathrm{DP}}}$ as a function of the number of steps rescaled by ${\left(p_{1}-p_{1}^{c}\right)}^{{\nu_{\|}}_{\mathrm{DP}}}$ for $p_{1}\in$ [0.260:0.002:0.280] surrounding the presumed critical value $p_{1}^{c}$, with the exponents corresponding to DP for D=1.

**Figure 12 entropy-22-01348-f012:**
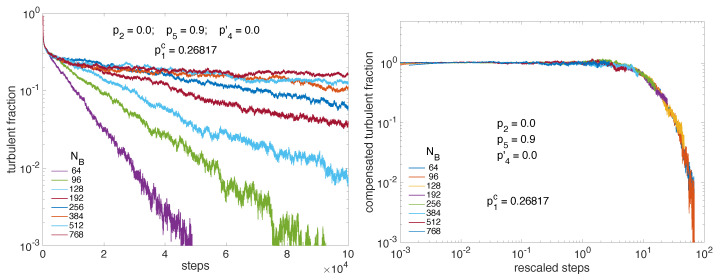
Size effects when $p_{2}=0$. (**Left**): Raw data showing late exponential decay and progressive prevalence of power-law decay as $N_{B}$ grows. (**Right**): Rescaled data. According to scaling theory, the appropriate scale for the number of steps (time *t*) is ${N_{B}}^{z/D}$; hence, $t\mapsto t/N_{B}^{z/D}$, with z≈1.58 when D=1, while the turbulent fraction has to be compensated for decay as $F_{t}\times t^{\alpha }$ with α=0.159. The collapse of traces illustrates universality with respect to 1D-DP.

**Figure 13 entropy-22-01348-f013:**
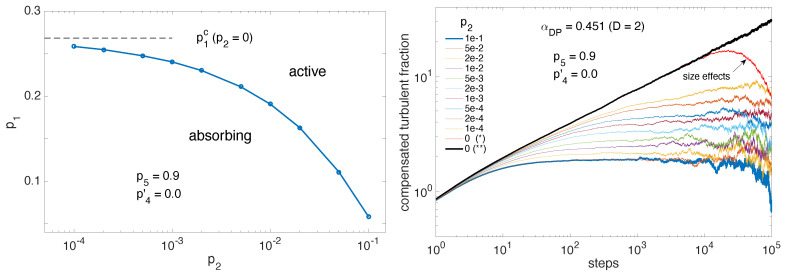
Crossover $p_{2}\to 0$. (**Left**): Criticality condition separating the sustained active regime from the absorbing regime. (**Right**): A DP-like process governs the decay when the line is crossed, the characteristics of which can be understood from the asymptotic power-law decrease of the turbulent fraction as a function of time, here compensated by $t^{\alpha }$, with $\alpha =\alpha_{\mathrm{DP}}\left(D=2\right)$.

**Figure 14 entropy-22-01348-f014:**
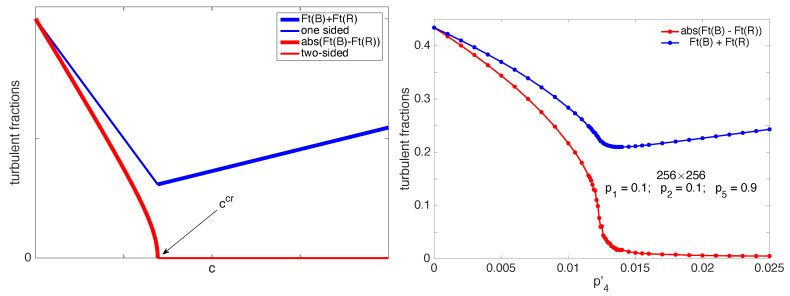
(**Left**): Bifurcation diagram of system (Equation 11) and (Equation 12) after [14]. The total turbulent fraction is $F_{t}\left(B\right)+F_{t}\left(R\right)$ and the order parameter characterizing the transition is $\mathrm{abs}(F_{t}\left(B\right)-F_{t}\left(R\right))$. A standard supercritical bifurcation is expected for this quantity with $\mathrm{abs}\left(F_{t}\left(B\right)-F_{t}\left(R\right)\right)\propto {\left(c^{c}-c\right)}^{1/2}$ in the one-sided regime, whereas $F_{t}\left(B\right)=F_{t}\left(R\right)$ in the two-sided regime. (**Right**): Time average of turbulent fractions as functions of the control parameter $p_{4}^{\prime }$ after elimination of an appropriate transient as obtained from simulations of the stochastic model.

**Figure 15 entropy-22-01348-f015:**
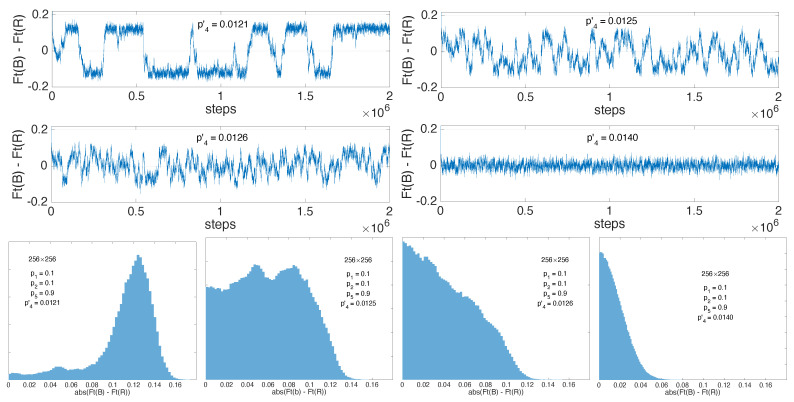
Time series of the instantaneous mean orientation as measured by $F_{t}\left(B\right)-F_{t}\left(R\right)$ in a 256×256 domain for $p_{1}=p_{2}=0.1$ and $p_{5}=0.9$, at $p_{4}^{\prime }=0.0121$ below the onset of the one-sided regime, at $p_{4}^{\prime }=0.0125$ and 0.0126 near the bifurcation point, and at $p_{4}^{\prime }=0.0140$ in the two-sided regime. Bottom line: Corresponding histograms of |Ft(B)−Ft(R)|. The histograms were all built using 75 bins and contain the same number of points for $10^{5}<t<2\times 10^{6}$, but the vertical scales are not identical.

**Figure 16 entropy-22-01348-f016:**
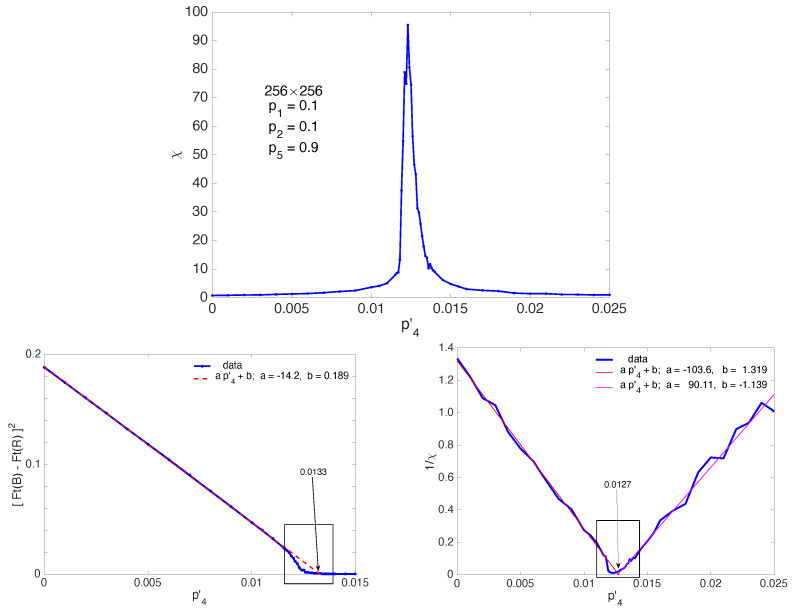
(**Top**): Variation of the susceptibility χ as a function of $p_{4}^{\prime }$. (**Bottom**): Evidence of mean-field behavior away from the critical regime: The order parameter squared (**left**) and the inverse of the susceptibility (**right**) both vary linearly with $p_{4}^{\prime }$ sufficiently far from the mean-field extrapolated threshold.

**Table 1 entropy-22-01348-t001:** Values of $p_{1}$ at criticality at given $p_{2}$ ($p_{5}=0.9$ and $p_{4}^{\prime }=0$).

*p* _2_	0.0	0.0001	0.0002	0.0005	0.001	0.002	0.005	0.01	0.02	0.05	0.1
$p_{1}^{c}$	0.2682	0.2585	0.2548	0.2476	0.2404	0.2302	0.2111	0.1907	0.1629	0.1109	0.0584

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
