# Peer review of "Transitional Channel Flow: A Minimal Stochastic Model"

_entropy, 2020, doi:10.3390/e22121348_

Round 1

Reviewer 1 Report

This paper presents a mimimal stochastic model for the transitional channel flow. In general, the study is well conducted and written. It definitely brings new results and insights on the problem. Therefore, the reviewer strongly recommend its publication as it is.

Author Response

We thank the Referee for her/his kind appreciation of our work.

Best regards.

Reviewer 2 Report

The manuscript constructed a PCA as a model to understand the stochastic evolution of the localized turblent bands (LTB) in transitional channel flows. The simulation results were shown to display the scaling behaviours of directed percolation at the critical points, which has been shown previously to be a plausible model for the evolution of the LTB population. The dependence of the scaling behaviours on a number of transitional probabilities is discussed and shown to be consistent with observations in transitional channel flows.

The results are interesting. I'm happy to accept the manuscript for publication. A few comments are as follows:

  1. Figure 1: 'prevail up the threshold'. should be 'up to'? Also, no need for capitalisation for "Channel"
  2. Line 73: 'upper tight'. Typo?
  3. Figure 6: 'depicted in Fig6(a)'. Might that be Fig5(a)?
  4. Line 312: 'support the neglect the'. Typo?
  5. Between line 327 and 328: 'the model can manage'. What does 'manage' mean here? 'for which with p4'=0'. 'with' is not necessary.
  6. Line 360: How Ft(0) is defined?
  7. Line 382: 'specie' is a typo
  8. Line 392: 'Ft=1'. How is Ft defined?
  9. Figure 8: The meaning of 'Ft(one sided, ...)' in the legend is not clear to me.
  10. Section 5: to make it easier for the readers who want to have a quick understanding of the main conclusions, I would suggest avoid using abbreviations (LTB, PCA, DP etc). Also, please explain in physical terms the meanings of p4', p2, p5 etc. Line 672: 'the two first'?

Author Response

We thank the Referee for her/his kind appreciation of our work and careful examination of the text.
Typos have been corrected:

  • point 1, OK;
  • point 2: l.73, 'upper' is suppressed because the 'tight' regime is well-distinguished from the lower 'loose' regime in Fig.1);
  • point 3, OK;
  • point 4, on l.312, 'supports the neglect the contribution of configurations…' has been corrected into 'support neglecting the contribution of configurations…';
  • point 5, 'manage' has been changed into 'deal with' and 'for which with p'_4 = 0' has been corrected by suppressing 'with' as recommended;
  •  point 7, two other occurrences of 'specie' have been corrected in addition to that on l.282;

For point 6, concerning the non-labelled paragraph between l.327 and 328, since we are speaking of F_t(B) and F_t(R) we think that the mention of 'F_t(0)' by the Referee might refer to F_t(\emptyset), the fraction of absorbing states. However, we are only interested in the fraction of active sites, and there is no need to define any absorbing fraction.

About points 6, 8, 9, otherwise, we found mentions of 'F_t(0)' only on line 392 and in the inset legend of Fig.8 (top). Here and there, '0' was meant for 't = 0'. So, the confusing part of the legend in Fig.8 (top) has been suppressed and the caption modified accordingly. Other slight cosmetic modifications have been made in the main text, in particular by moving 386-387 from the end of a paragraph to the head of the next one and dropping the explicit reference to time t = 0 as in the form 'F_t(0)' pointed out by the Referee.

All abbreviations have been suppressed from §5 as requested. A brief reminder of the physical meaning of p'_4, p_2 and p_5 is now given.

We hope having revised the text following the Referee's remarks.
Best regards.